# Evolutionary and Gene Expression Analyses Reveal New Insights into the Role of *LSU* Gene-Family in Plant Responses to Sulfate-Deficiency

**DOI:** 10.3390/plants11121526

**Published:** 2022-06-07

**Authors:** Felipe Uribe, Carlos Henríquez-Valencia, Anita Arenas-M, Joaquín Medina, Elena A. Vidal, Javier Canales

**Affiliations:** 1Instituto de Bioquímica y Microbiología, Facultad de Ciencias, Universidad Austral de Chile, Valdivia 5110566, Chile; f.uribecardenas@gmail.com (F.U.); carlos_henriquezva@hotmail.com (C.H.-V.); anitamaribel@gmail.com (A.A.-M.); 2ANID-Millennium Science Initiative Program-Millennium Institute for Integrative Biology (iBio), Santiago 8331150, Chile; elena.vidal@umayor.cl; 3Centro de Biotecnología y Genómica de Plantas, INIA-CSIC-Universidad Politécnica de Madrid, 28223 Madrid, Spain; medina.joaquin@inia.es; 4Centro de Genómica y Bioinformática, Facultad de Ciencias, Universidad Mayor, Santiago 8580745, Chile; 5Escuela de Biotecnología, Facultad de Ciencias, Universidad Mayor, Santiago 8580745, Chile

**Keywords:** sulfate deficiency, *Arabidopsis thaliana*, *Solanum lycopersicum*, *Triticum aestivum*, LSU, response to low sulfur, abiotic stress, sulfur nutrition

## Abstract

*LSU* proteins belong to a plant-specific gene family initially characterized by their strong induction in response to sulfate (S) deficiency. In the last few years, LSUs have arisen as relevant hubs in protein–protein interaction networks, in which they play relevant roles in the response to abiotic and biotic stresses. Most of our knowledge on LSU genomic organization, expression and function comes from studies in Arabidopsis and tobacco, while little is known about the *LSU* gene repertoire and evolution of this family in land plants. In this work, a total of 270 LSU family members were identified using 134 land plant species with whole-genome sequences available. Phylogenetic analysis revealed that *LSU* genes belong to a *Spermatophyta*-specific gene family, and their homologs are distributed in three major groups, two for dicotyledons and one group for monocotyledons. Protein sequence analyses showed four new motifs that further support the subgroup classification by phylogenetic analyses. Moreover, we analyzed the expression of *LSU* genes in one representative species of each phylogenetic group (wheat, tomato and Arabidopsis) and found a conserved response to S deficiency, suggesting that these genes might play a key role in S stress responses. In summary, our results indicate that *LSU* genes belong to the *Spermatophyta*-specific gene family and their response to S deficiency is conserved in angiosperms.

## 1. Introduction

Sulfur is an essential macronutrient for plants and a constituent of relevant biomolecules, such as the amino acids methionine and cysteine, the antioxidant glutathione, glucosinolates, coenzymes and prosthetic groups [1]. As such, the availability of S, the main source of sulfur in soils, is an important determinant of plant growth, yield and quality.

In recent years, S deficiency has become widespread in many regions of the world, mainly due to anthropogenic factors such as the use of fertilizers with a diminished content of S, intensive agriculture, diminished use of sulfur-containing fungicides and the advent of environmental policies that limit the emissions of SO_2_ [2]. Due to the inability of animals to synthesize S-containing amino acids, plant-derived sulfur-containing compounds are key for the nutrition and survival of humans and livestock [3]. Thus, understanding how plants cope with S deficiency is of relevance for improving S content and the productivity of crops.

Plant responses to S deficiency have been studied both at the physiological and molecular levels. At the physiological level, S deficiency leads to reduced sulfur content and reduced metabolic activity, alters photosynthetic activity and produces oxidative stress due to an accumulation of reactive oxygen species. This leads to plants with stunted growth and leaf chlorosis [4]. At the molecular level, several S deficiency-responsive genes have been identified using omics approaches, indicating a massive reprogramming of gene expression in response to the availability of this nutrient [5,6,7,8,9,10,11,12,13,14]. A meta-analysis of published transcriptomics data of the S response in Arabidopsis identified genes whose expression is consistently controlled by S deficiency [15], including S transporters and enzymes related to S assimilation, as well as genes participating in cell wall organization, regulation of proteolysis and C/N metabolism. Among the consistently induced genes were *LSU1* and *LSU2*, members of the plant-specific “RESPONSE TO LOW SULFUR (LSU)” family, composed of four members in Arabidopsis (LSU1-4). *LSU1* and *LSU2* were first identified as strongly induced at the transcript level by S deficiency in Arabidopsis [6,16] as well as at the protein level [17]. The analysis of LSU-GFP transgenic lines in Arabidopsis shows that LSU1 is preferentially located in chloroplasts of guard cells and LSU2 in the nucleus of epidermal root cells [17]. Another study showed that LSU2 protein is also localized in chloroplasts [18], and cytosolic localization has also been reported for LSU1 and LSU2 in Arabidopsis [17]. Similar to LSU2, tobacco LSU homolog UP9C is localized in the nucleus of root cells [16]. UP9C has been described as necessary for tobacco response to S deficiency and possesses a 20-nt DNA motif in its promoter, termed UPE-box, that confers S-deficiency responsiveness [19]. This element is also present in the promoters of Arabidopsis *LSU1*, *LSU2* and *LSU3*, as well as in promoters from other S-responsive genes. such as *APS reductase 1* (*APR1*), *APR3* and S-transporter *SULTR2;1* [20]. Mutations within this motif affect the binding of the NtEIL2 transcription factor, as well as the binding of its Arabidopsis homolog SLIM1 [19], a key transcription factor mediating the S-deficiency response [21], further supporting a role for LSU proteins in response to this nutrient.

LSUs are small proteins of approximately 90–100 amino acids. Molecular modeling of LSU structures in Arabidopsis shows that these small proteins have a coiled-coil structure [22]. Consistently, circular dichroism analysis of a recombinant LSU protein (UP9C) indicates that this protein is almost fully alpha-helical [16], which suggests that LSUs are flexible and might be able to bind other proteins. Accordingly, LSUs appear as hubs in protein–protein interaction networks in Arabidopsis [22,23,24,25]. A recent study also showed that LSU proteins are able to generate multimers, forming hetero and homodimers [22].

Besides S deficiency, LSU transcript levels change in response to other abiotic stresses, such as H_2_O_2_ treatment [26], C-starvation [27], treatment with lincomycin, an inhibitor of chloroplast biogenesis [28], iron deficiency, copper excess, salt stress or high pH [17]. LSUs have also been linked to the plant response to biotic stress. Interactome analysis in Arabidopsis has shown that LSUs are targeted by effectors from different pathogens, including the bacterium *Pseudomonas syringae*, the oomycete *Hyalopernosopora arabidopsidis* and the fungus *Golovinomyces orontii* [24,29]. Furthermore, mutant *lsu2* plants present an enhanced susceptibility to infection by these pathogens, while *LSU2* overexpression generates a resistant phenotype [24,29]. Consistent with the role of LSUs in plant defense, LSU1 has been shown to bind Fe superoxide dismutase 2 (FSD2), increasing its enzymatic activity and leading to enhanced chloroplastic H_2_O_2_ production, an important element of pattern-triggered immunity. The binding of pathogen effectors to LSU1 could interfere with this process, leading to plant susceptibility [17]. On the other hand, infection by *Pseudomonas syringae* leads to a disruption of the physical interaction between LSU2 and the AtRAP protein in the chloroplast, positively regulating host defense [18]. Interestingly, LSU downregulation results in a higher susceptibility to *Pseudomonas syringae* infection upon S deficiency, indicating that LSU proteins mediate defense responses during this nutritional stress [17]. Consistently, LSU1–4 are able to interact in vivo with the S assimilation enzyme ATP sulfurylase (APS1) in Arabidopsis [22], suggesting a potential direct regulation of S metabolism by LSUs. Furthermore, evidence points to a possible role of LSU-like proteins as common regulatory elements of the S and ethylene signaling pathways. UPC9 is able to interact with ACC oxidase, and silencing of *UP9C* leads to a reduction in ethylene biosynthesis during S-deficiency [30]. Additionally, LSU2 is directly regulated by EIN3 [31], a key transcription factor of ethylene signaling.

In terms of genomic organization, LSUs in Arabidopsis are located in pairs of directed gene repeats (*LSU1* and *LSU3* in chromosome 3 and *LSU2* and *LSU4* in chromosome 5) [32], indicating a possible gene duplication event. In tobacco, six LSU homologs have been described (*UP9A-F*). LSU homologs can be found by BLAST search across all higher land plants [17,20], and protein alignment of LSUs from different plants shows a short, conserved region of yet unidentified function [32]. However, no systematic identification and characterization of *LSU* genes in plants have been performed to date. Therefore, the origin and the evolution of *LSU* genes in plants remain unknown. On the other hand, studies on the evolutionary history of genes involved in S transport and metabolism [33,34] have shown that all genes of the S assimilation pathway of higher plants can be found in *Chlamydomonas reinhardtii*, *Physcomitrella patens* and *Selaginella moellendorffii* genomes [33,34]. However, significant differences were found in the genomic organization of these genes in basal land plants, showing a reduced complexity in some aspects of sulfur metabolism, such as cysteine synthesis or the synthesis of sulfated metabolites [34].

In this work, we performed a genome-wide identification and evolutionary analysis of *LSU* genes in plants. Our results suggest that different genes from the S assimilation pathway, the LSU gene family, arose in *Spermatophyta*. Phylogenetic analysis revealed three major LSU groups in angiosperms, two for dicotyledons and one group for monocotyledons. Moreover, we analyzed the expression of *LSUs* in one representative species of each phylogenetic group and found a conserved response to S deficiency, suggesting that these genes might play a key role in sulfur-stress responses. Overall, these data indicate that *LSU* genes are evolutionarily conserved in angiosperms, and members of this family might play a significant role in the regulation of S transport and assimilation.

## 2. Results

### 2.1. Genome-Wide Identification of LSU Genes in Plants Reveals That LSUs Are a Spermatophyta-Specific Gene Family

We performed a genome-wide search of *LSU* genes in the PLAZA 5.0 database [35] as the first step towards a phylogenetic analysis of this gene family in plants. This database contains structural and functional annotations of 134 high-quality plant genomes, including angiosperms, gymnosperms, non-seed plants and microalgae [35] (Figure 1A). We used the *LSU* genes of *Arabidopsis thaliana* as a reference to identify 270 genes of this family in the PLAZA 5.0 database (Appendix A). Interestingly, all *LSU* genes belong to angiosperm and gymnosperm species, suggesting that *LSU*s are a *Spermatophyta*-specific gene family (Figure 1A). Accordingly, these homolog groups are annotated as an exclusively *Spermatophyte* family in PLAZA 5.0.

*LSU1/LSU3* and *LSU2/LSU4* genes in Arabidopsis are localized close to each other in chromosomes 3 and 5, respectively, suggesting that each of these pairs of *LSU* genes arose from a duplication event [32]. To investigate the potential expansion of the LSU family in plants, we analyzed the number of genes of this family across angiosperm genomes. The number of *LSU* genes is variable, ranging from 1 to 9 members (Appendix A). In general, the species with a higher number of *LSU* genes were polyploids, so we performed a correlation analysis to verify the statistical significance of this observation. We found a significant positive correlation between the *LSU* gene number and the total number of coding genes (Appendix A), suggesting that the variation of gene copy number in the *LSU* family depends on genome size. Considering this correlation, we normalized the *LSU* copy number by the total number of coding genes to compare major taxonomic groups of angiosperms. As shown in Figure 1B, the distribution of the normalized number of *LSU* genes is very similar in monocotyledons and eudicotyledons species, with an average number of 2 *LSU* genes per 34.075 coding genes. Indeed, no significant differences in the number of *LSU* sequences were obtained between these taxonomic groups (*p*-value = 0.2; Appendix A). Therefore, these results suggest no evolutionary trend toward expansion or reduction of this family in angiosperm plants.

### 2.2. Phylogenetics Analysis of the LSU Gene Family

To reveal the evolutionary relationships of the 270 *LSU* genes identified in the PLAZA 5.0 database [35] among the seed-plant lineages, we constructed a phylogenetic tree using the maximum likelihood method implemented in IQ-TREE 2 [37] with the LSU protein sequences. The tree showed that these LSU members can be divided into three major phylogenetic groups: Group A, including most of the monocotyledon species; Group B, including most of the malvid species and Group C, including most of the rosid species (Figure 2). This organization of *LSU* genes fits very well with the reported phylogeny of angiosperm plants [38]. One important exception was the case of basal angiosperms, which are clustered together with the species of Group C; however, the bootstrap support for this case is very low (aLRT < 70%).

The phylogenetic analysis also showed that all *LSU* members of the same species in Groups B and C are clustered very close, suggesting a low intraspecific sequence divergence in the species of these groups. In contrast, most *LSU* genes of the same species in Group A are separated into two different clusters (Figure 2). To elaborate on this observation, we computed the evolutionary distance between each pair of sequences of the same species using MEGA 11 with the JTT matrix-based model [39]. As shown in Figure 3A, the intraspecific distance of *LSU* genes in Group A is significantly higher (*p*-value < 0.0001) than those of Groups B and C. Then, we selected one species per group to visualize these differences more clearly. We selected bread wheat (*Triticum aestivum*) for Group A, whose *LSU* genes cluster into two homoeolog groups: three on chromosome 1 (*TraesCS1A03G0484600*, *TraesCS1B03G0587700*, *TraesCS1D03G0456700*) and two on chromosome 6 (*TraesCS6A03G0192500*, *TraesCS6B03G0289700*). As shown in Figure 3B, the percent of amino acid identity is approximately 37–39% between the two homoeolog groups, indicating a high degree of divergence between members of this family in wheat. In contrast, for Arabidopsis in Group B, the lower percentage of identity between LSUs is 47–50%, with most of the LSUs sharing 60% or more of identity. The same trend is seen for tomato in Group C, where most LSUs share more than 70% of identity (Figure 3B). These results suggest significant differences in the intraspecific evolution of *LSU* genes between monocotyledons and eudicotyledons.

Multiple sequence alignment revealed that LSU proteins of the selected species exhibited significant differences in the N-terminal and C-terminal regions (Figure 3C). In contrast, the central region of LSU proteins showed two highly conserved domains, one from the amino acid positions 31 to 49 and the other one from 59 to 95 (Figure 3C). To extend this analysis to the whole 270 LSU proteins and reveal the structural variation of *LSU* genes in angiosperms, we predicted putative motifs using the MEME tool from the MEME Suite [40]. As shown in Figure 4A, we found a total of five distinct motifs with different degrees of distribution between phylogenetic groups. Motifs 1 and 2 were widely present in most members (>90%) of the three phylogenetic groups, and these conserved motifs were found in the central region of LSU proteins (Figure 4B). It is important to note that motif 1 matches with the only reported motif of the LSU family (A-x-x-x-E-E-x-L-C-x-x-L-x-[E/D]-x-[E/D]) [22]. In contrast, motif 3 was present in the C-terminal regions of LSU proteins and showed a limited distribution in Group A compared to Groups B or C (62% versus 76–89%). The motifs found in the N-terminal regions (motif 4 and motif 5) also showed different degrees of distribution between phylogenetic groups (Figure 4A). These results suggest that the amino acid residues outside of motifs 1 and 2 have significantly changed during plant evolution.

### 2.3. Expression Analyses of the LSU Family in Wheat, Tomato and Arabidopsis Reveal a Conserved Response to S-Deficiency

To elucidate if the response of *LSU* genes to S deficiency is conserved in angiosperms, one species of each phylogenetic group was selected for mRNA expression analysis: wheat from Group A, Arabidopsis from Group B and tomato from Group C. We grew plants in liquid medium in a complete medium (control condition) or a medium lacking S (S-deficiency condition), as previously described [14], until the emergence of the second true leaves. In this growth stage, transcript levels in roots and leaves samples were analyzed by reverse transcription-quantitative real-time polymerase chain reaction (RT-qPCR). First, we evaluated the mRNA levels of classical S-responsive genes as a positive control of the nutritional treatment in each species (Appendix A). In Arabidopsis and tomato plants, a significant increase in mRNA levels of *SULTR1.2* and *APR2* genes, encoding an S transporter and an S assimilation enzyme, respectively, was observed in both organs in response to S deficiency. In the case of wheat, we analyzed the orthologs of Arabidopsis *MORE SULPHUR ACCUMULATION1* (*MSA1*) and APR2, which also showed higher expression levels under S deficiency in both root and leaves (*p*-value < 0.05; Appendix A). These results indicate that the S-deficiency response is triggered at the selected developmental stage in all species.

In the case of Arabidopsis, all *LSU* genes showed a significant increase in mRNA levels under S deficiency (*p*-value < 0.05) in root and leaves, with *LSU4* showing a lower expression than the other *LSU* genes in all conditions (Figure 5). A similar result was observed in tomato plants, in which all *LSU* genes showed increased mRNA levels under S deficiency in both roots and leaves, with one of them (*Solyc03g096770.1*) showing a lower expression in all samples (Figure 5).

In the case of wheat, we analyzed the expression of the two homoeolog groups, *TaLSU1* corresponding to homoeologs of chromosome 1 and *TaLSU2* in the case of homoeologs of chromosome 6. As shown in Figure 5, we found a significant induction of *TaLSU1* by S deficiency in both organs. On the other hand, the mRNA levels of *TaLSU2* were very low in all analyzed samples; this could be due to a developmental effect since public data from the Wheat eFP Browser [43] shows that the expression of this gene is very low in the seedling stage (Appendix A).

Arabidopsis and tobacco *LSU* genes contain a cis-regulatory element termed the UPE-box, which is responsible for the S-deficiency response of *LSU1* and other S-responsive genes [19,44]. To investigate whether the UPE-box is conserved across promoters of the *LSU* genes analyzed by RT-qPCR in this study, we searched for this cis-regulatory element in the 2Kb upstream regions of the transcription start site of each gene using the FIMO tool [45]. As shown in Figure 6, all *LSU* genes showing a significant response to S-deficiency have at least 3 UPE-box elements in their promoter region. In the case of tomato LSUs, the Arabidopsis *LSU1* and the wheat *TaLSU1* gene, the UPE-box are located near the transcriptional start site. On the other hand, previous reports have shown that the transcription factor SLIM1 binds to the *LSU1* promoter through the UPE-box element [19]. To investigate whether ortholog genes of *SLIM1* are present in wheat and tomato genomes, we performed a genome-wide search in the PLAZA 5.0 database [35]. We found that *SLIM1* belongs to the homologous gene family HOM05D000843 with 35 genes in the three selected species. Phylogenetic analysis showed that tomato has two *SLIM1* orthologs genes, similar to wheat, which has two homoeolog groups (Figure 6B).

## 3. Discussion

### 3.1. Molecular Evolution of the LSU Family

In this work, we performed an evolutionary analysis of the *LSU* family in plants using the 134 high-quality genomes available in the PLAZA 5.0 database. We found *LSU* homolog sequences in gymnosperm and angiosperm species, indicating that *LSU* genes are a *Spermatophyta*-specific gene family. However, typical S-responsive genes, such as S transporters, or genes encoding enzymes of S assimilation, such as *ATP sulfurylase* (*ATPS*) or *APS reductase* (*APR*), are present in all Viridiplantae from microalgae to angiosperms [34], indicating that the evolutionary appearance of the *LSU* family is recent compared to S assimilation genes. Interestingly, several experimentally verified interactors of *LSU* genes in Arabidopsis, such as *APS1*, *GAPC1*, *RAF2*, *FSD2* and *RAP1* [17,18,22], are also present in all analyzed Viridiplantae genomes (HOM05D001870, HOM05D000557, HOM05D002852, HOM05D002348 and HOM05D006787 homologous gene family, respectively, in PLAZA 5.0), suggesting that LSU-target regulatory interactions do not require the same evolutionary history.

A recent study analyzed the evolution of land plants using 208 complete genomes [49] and identified 1432 *Spermatophyta*-specific genes. GO ontology enrichment analysis revealed that *Spermatophyta*-specific genes are over-represented in GO terms associated with developmental processes and phytohormone responses. In this regard, several lines of evidence have related *LSU* genes and phytohormones [30,31]. For instance, it has been proposed that *LSU* genes may establish a communication between the phytohormone ethylene and S assimilation in response to cadmium stress [31]. Therefore, *LSU* genes may be related to phytohormone responses, similar to other *Spermatophyta*-specific genes.

The analysis of *LSU* sequences across angiosperms also reveals that the evolutionary distance between *LSU* genes of the same species in monocotyledons is significantly higher than in eudicotyledons, suggesting a potential functional divergence of *LSU* genes within monocotyledon species, such as wheat. In plants, functional divergence is also reflected in contrasting expression profiles of divergent genes due to the well-known correlation between gene expression and function in plants [50,51,52]. In fact, the expression patterns of the two *LSU* genes with a lower percentage of sequence identity in wheat (*TraesCS1A03G0484600* and *TraesCS6A03G0192500*) show important differences during development according to the WheatOmics database [53]. *TraesCS1A03G0484600* is preferentially expressed in roots and spikes, whereas *TraesCS6A03G0192500* shows higher transcript levels in leaves and stems at the flowering stage (Appendix A). Consistent with these expression patterns, we found that the expression of *TraesCS1A03G0484600* was significantly higher than *TraesCS6A03G0192500* in all analyzed samples of the early seedling stage.

Previous reports have shown that the LSU proteins have a small conserved motif of 16 amino acids (A-x-x-x-E-E-x-L-C-x-x-L-x-[E/D]-x-[E/D]) [32]. However, this motif was deduced from a reduced set of sequences (42 protein sequences from 15 species) and, therefore, other relevant domains in the evolution of these proteins could be identified by expanding this analysis to other plant species. Indeed, our conserved motif analysis identified four new putative motifs based on 270 LSU protein sequences from 112 species (Figure 4). Interestingly, one of these new motifs (motif 2) also showed a high degree of conservation, suggesting that LSU proteins might have two essential domains. Our results also suggest that the known motif is of a much greater extent than previously reported [32] (from 16 to 29 amino acids). The significance of these strongly evolutionarily conserved regions is unknown, but a recent study suggests that particular motifs in LSU proteins are not required for the interaction of LSU with other molecular targets [22].

### 3.2. Expression Profiling Unveils a Conserved Function in the Control of S Responses

Previous studies have shown that *LSU1-3* of Arabidopsis strongly responds to S deficiency as well as the *LSU* genes of tobacco plants [32]. In this work, we extended these expression analyses to wheat and tomato plants using the same experimental design and similar developmental stages. We found that all *LSU* genes were induced in response to S deficiency in roots and leaves of Arabidopsis and tomato plants (Figure 3), suggesting a functional redundancy in response to this nutritional stress. Accordingly, a recent study of the interactome of LSU proteins in Arabidopsis showed similar binding properties of LSU1-4 [22]. On the other hand, the fold change and expression levels of *LSU4*, *TaLSU2* and *Solyc03g096770.1* were remarkably lower than the other *LSU* genes in our experimental conditions. These exceptions may be due to a different temporal response to S deficiency or to a developmental effect. This last hypothesis is consistent with the developmental expression pattern of the *LSU4* gene in Arabidopsis, which is preferentially expressed in flowers, according to the ePlant database [54]. Furthermore, a mutation in the *LSU4* gene is known to affect flower development in Arabidopsis [55]. A similar case of tissue-specific expression of *LSU* genes occurs with wheat, as discussed below. These findings suggest that some members of the *LSU* family may have a tissue-specific response to S deficiency.

Cross-species genome-wide expression analysis showed a significant induction by S deficiency of at least one *LSU* gene member in all analyzed plant species, suggesting that they might play a conserved role in response to S stress conditions in angiosperms. It is important to note that the selected species belong to the three major *LSU* clades detected in the phylogenetic analysis (Figure 2). Moreover, the induction of *LSU* genes in tobacco plants has been also reported [16]. This conserved response to S availability strongly suggests a functional role of LSUs in this nutritional response. Interestingly, TAP-MS approaches in Arabidopsis have identified ATP sulfurylase 1 (APS1) is a protein directly interacting with LSU1, and this interaction has also been validated in planta BiFC assays [22].

The conserved response to S deficiency of the *LSU* genes analyzed in this work also suggests functional conservation of cis-regulatory elements in Arabidopsis, tomato and wheat plants. Consistently, the UPE-box cis-regulatory element was found in the promoter regions of *LSU* genes of these species (Figure 6A). UPE-box is a cis-regulatory element involved in the S-response that is present in the promoters of several S deficiency-responsive genes, including *LSU1* [19,44]. Moreover, it has been demonstrated that the mutation of the UPE box affects the yeast-one-hybrid binding strength of SLIM1 to the promoter of *LSU1* [19,44]. Interestingly, we found that wheat and tomato plants have two ortholog genes to the Arabidopsis SLIM1, suggesting a possible conserved regulatory mechanism to the S deficiency. In addition, we previously reported that one of the ortholog genes of SLIM1 in tomato (*Solyc01g006650*) has a significant response to S deficiency in roots and leaves [14]. The predicted target genes of this TF in tomato include classical S-responsive genes, such as LSUs, and genes involved in S assimilation [14], supporting the idea of a conserved regulatory mechanism for the S deficiency in Arabidopsis, tomato and wheat plants.

In summary, we found that *LSU* genes belong to a *Spermatophyta*-specific gene family and their response to S deficiency is conserved in angiosperms. Our analyses also suggest that homolog genes to the Arabidopsis SLIM1 might be involved in this conserved nutritional response. Further research should be undertaken in other angiosperm species, such as wheat or tomato, to verify this potential role of the transcription factor orthologs to *SLIM1* in regulating S-deficiency responses.

## 4. Materials and Methods

### 4.1. Identification of LSU Genes and Phylogenetic Analysis

The sequence and gene codes of Arabidopsis LSU proteins were retrieved from the Arabidopsis Information Resource (https://www.arabidopsis.org/; accessed on 6 September 2021). *LSU* genes of other plant species were identified by BLASTP searches across the set of protein sequences obtained from the 134 species publicly available in the PLAZA 5.0 database (http://bioinformatics.psb.ugent.be/plaza/; accessed on 13 September 2021) [35]. In this manner, we found a total of 270 LSU sequences corresponding to the PLAZA homologous gene family HOM05D003381, HOM05M005606 and HOM05M009560 (Appendix A).

Multiple sequence alignments were performed using the MAFFT software [41], and the alignments were visualized with Jalview 2 [42]. To identify and retain parsimony-informative sites, the alignments were trimmed using ClipKIT version 1.3.0 with default parameters [56]. IQ-TREE 2 [37] was used to perform phylogenetic analyses for maximum likelihood (ML) trees using the trimmed multiple alignments. The implemented ModelFinder function in IQ-TREE 2 software determined Q.plant + R5 amino acid replacement matrix [57] to be the best substitution model for tree inference. A total of 10,000 bootstrap replicates were performed using aLRT [58] and UFBoot tests [59] implemented in IQ-TREE 2 [37] for support estimation of reconstructed branches. This bioinformatic pipeline was also used for the case of SLIM1 phylogenetic analysis. The evolutionary distance between each pair of LSU sequences was estimated using MEGA 11 with the JTT matrix-based model [39].

### 4.2. Motif Analysis

Conserved protein motifs of the deduced LSU protein sequences were detected using the MEME tool from the MEME Suite v.5.4.1 [40]. The number of potential motifs was set up to 10 with having an optimum motif width ranging between 11 and 50 residues. The distribution of conserved motifs was visualized using TBtools v.0.665 [60].

The identification of UPE-box motif (AG[G/A]T[T/A]CATTGAA[T/C]CT[A/G]GAC[A/G]) in the promoter regions of *LSU* genes of Arabidopsis, tomato and wheat were performed using the FIMO tool from the MEME Suite v.5.4.1 [40] with a *p*-value threshold of 0.001. The promoter region was defined as the 2-kb sequence upstream of the transcription start site (based on Ensembl Plants v52) of each gene.

### 4.3. Plant Material and Growth Conditions

For the gene expression analysis of LSU gene family in response to S deficiency, we used seeds of the *Arabidopsis thaliana* ecotype Columbia (Col-0), *Solanum lycopersicum (cv. Moneymaker)* and *Triticum aestivum* L. in liquid medium with all nutrients available or lacking S, as we previously described [14], until the emergence of the second set of true leaves, which occurs 3 weeks after sowing for these plants under our experimental conditions.

### 4.4. RNA Extraction and RT-qPCR

Total RNA extraction was performed from 200 mg of frozen leaves or roots of Arabidopsis, wheat and tomato plants using the Spectrum Plant Total RNA kit (Sigma) and DNAase treatment with TURBO DNase (Invitrogen), according to the manufacturer’s protocols. cDNA was synthesized using 1000 ng of total RNA with All-In-One RT MasterMix 5X (abm). RT-qPCR reactions were performed with 25 ng of cDNA using the PowerUp SYBR Green Master Mix (Thermo Fisher Scientific) and CFX96 Touch Real-Time PCR Detection System (BioRad) according to the manufacturer’s instructions. The expression levels were normalized with reference genes that have proved stable in previous studies: the *ADAPTOR PROTEIN-4-MU-ADAPTIN* gene (AT4G24550) in the case of Arabidopsis [46], TIP4I-like family protein (TIP4I, SGN-U584254) for tomato [47] and the *Ubiquitin-conjugating enzyme* gene (*TraesCS4A02G414200*) for wheat [48]. Raw fluorescence data derived from each RT-qPCR reaction were analyzed using the Real-time PCR Miner 4.0 software to obtain Ct values and gene amplification efficiencies [61]. Sequences of the RT-qPCR primers used in this study are provided in Appendix A. In the case of wheat, conserved primers for each homoeolog group were designed (*TaLSU1* for *TraesCS1A03G0484600*, *TraesCS1B03G0587700*, *TraesCS1D03G0456700*; *TaLSU2* for *TraesCS6A03G0192500*, *TraesCS6B03G0289700*) as the sequences of *LSU* genes of all the respective homoeologs were almost the same in wheat sub-genomes. Moreover, we verified that *LSU* genes of the same homoeolog group have a significantly high correlation in public RNA-seq datasets (*p*-value < 0.05, Appendix A), indicating redundant expression profiles within the homoeolog group as previously reported for ~70% of wheat homoeolog triads [43].

## Figures and Tables

**Figure 1 plants-11-01526-f001:**
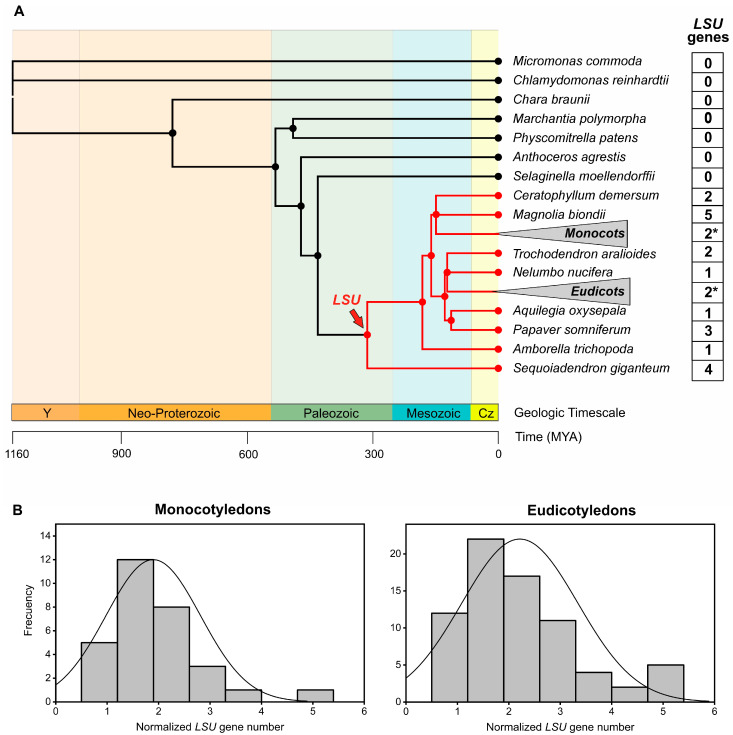
*LSUs* belong to a *Spermatophyta*-specific gene family. (**A**) The phylogeny of 134 plants from the PLAZA 5.0 database was used in this study. To improve visualization, 73 eudicotyledon and 31 monocotyledon species collapsed in the phylogenetic tree (triangle) and the average number of *LSU* genes are indicated with an asterisk. The order of tree branches and divergence time were obtained from the TimeTree database (http://timetree.org/; accessed on 13 December 2021) [36]. (**B**) The normalized number of *LSU* genes identified in angiosperms plants. The number of *LSU* genes of each species was divided by 34,075 coding genes, which is the average total number of genes in the analyzed species.

**Figure 2 plants-11-01526-f002:**
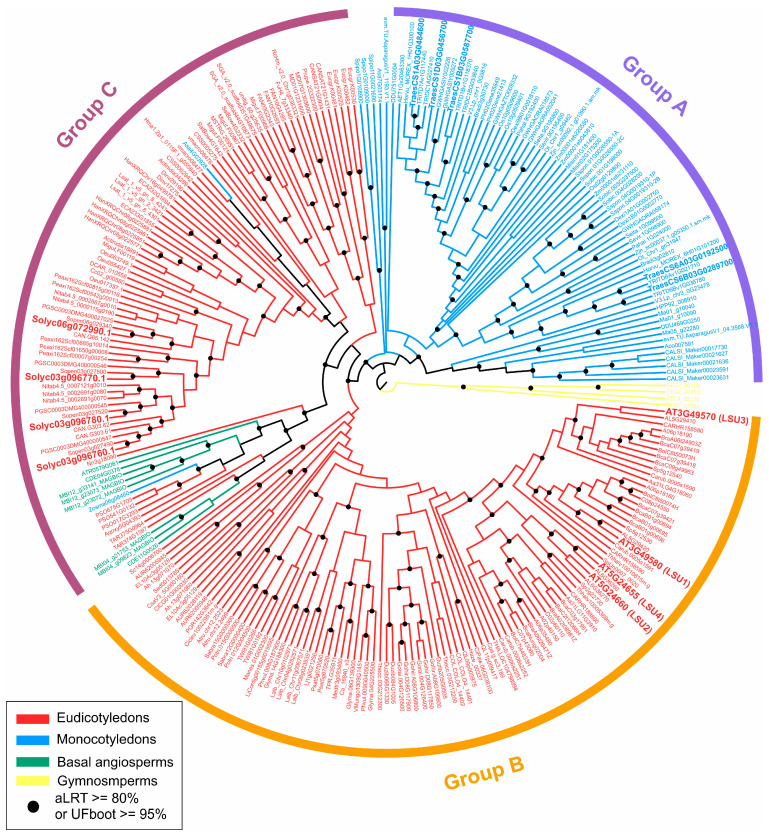
Phylogenetic analysis of *LSU* genes in angiosperm plant lineages. A phylogenetic tree was constructed using the maximum likelihood method from IQ-TREE 2 [37], supported by 10,000 bootstrap samples. Yellow lines represent gymnosperms, green lines represent basal angiosperms, blue lines represent monocotyledons and red lines represent eudicotyledons. The gene identifiers correspond to the PLAZA 5.0 database [35], and the complete list of *LSU* genes is shown in Appendix A.

**Figure 3 plants-11-01526-f003:**
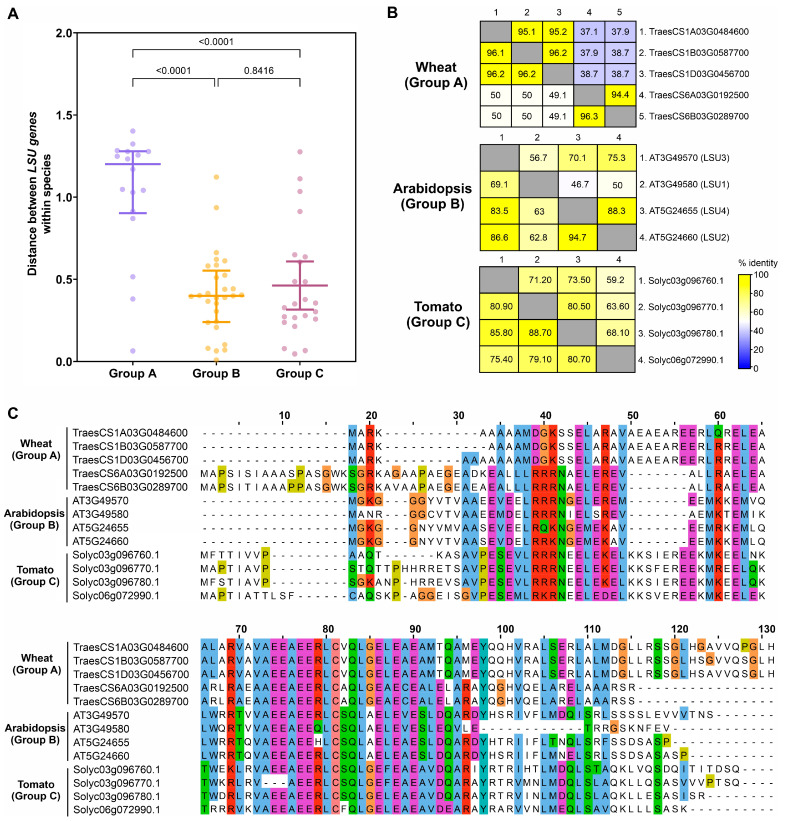
Comparative sequence analysis between representative species of the three phylogenetic groups of the LSU family. (**A**) The evolutionary distance between each pair of *LSU* sequences of the same species was obtained using MEGA 11 [39]. (**B**) The percentage of similarity and identity between every pair of *LSU* sequences within a representative species of each phylogenetic group. The upper part of the matrix represents the identity, and the lower part is the similarity. (**C**) Multiple sequence alignment of LSU proteins from wheat, tomato and Arabidopsis. Amino acid sequences were aligned using MAFFT [41] and visualized with Jalview 2 using the Clustal X color scheme [42].

**Figure 4 plants-11-01526-f004:**
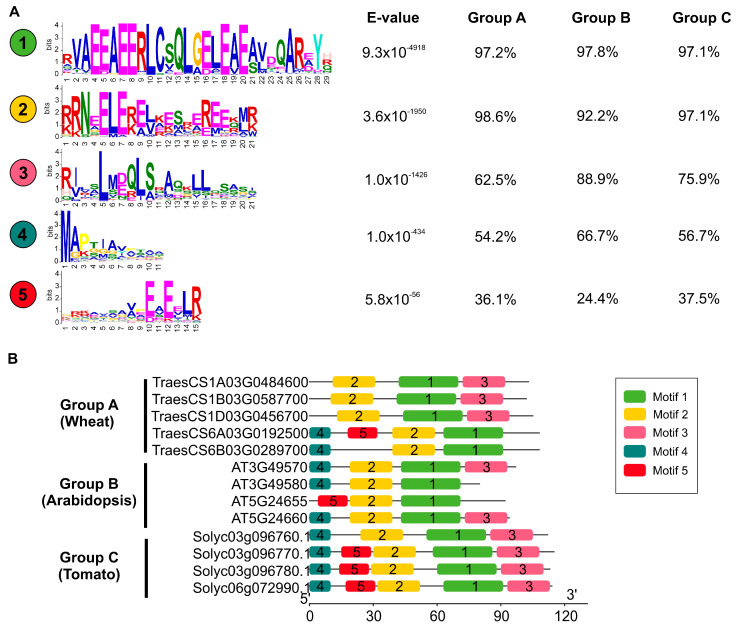
Conserved motif composition of the LSU gene family. (**A**) The conserved motifs were identified using the MEME tool [40], and the % distribution across the three phylogenetic groups is indicated right to the motif. (**B**) Five putative motifs are indicated by colored boxes in representative species of each phylogenetic group.

**Figure 5 plants-11-01526-f005:**
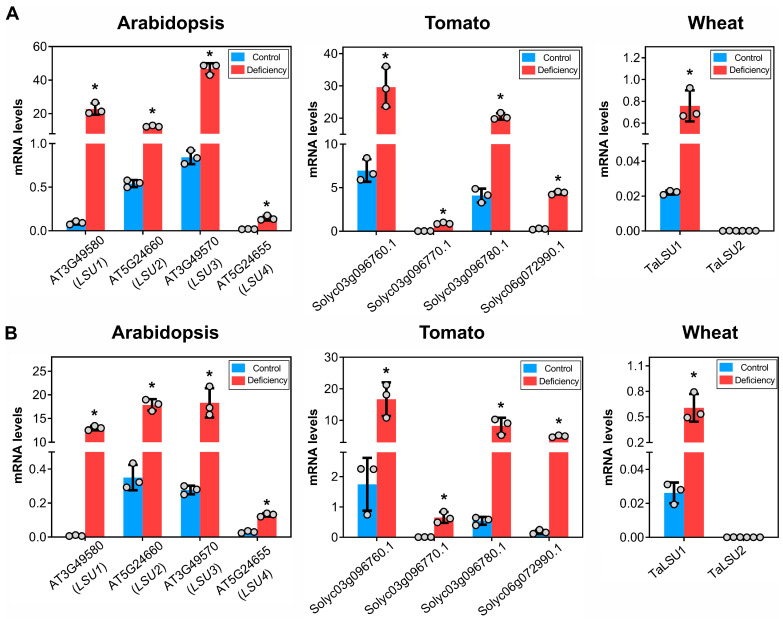
Expression profiles of *LSU* genes of wheat, Arabidopsis and tomato under S deficiency. The mRNA levels of the *LSU* gene family were measured by RT-qPCR in leaves (**A**) and roots (**B**) of each species grown for 3 weeks in a liquid medium under control and S-deficiency conditions. The expression levels were normalized with the *ADAPTOR PROTEIN-4-MU-ADAPTIN* gene (AT4G24550) in the case of Arabidopsis [46], TIP4I-like family protein (TIP4I, SGN-U584254) for tomato [47] and the Ubiquitin-conjugating enzyme gene (TraesCS4A02G414200) for wheat [48]. Each treatment was performed using three independent experiments. The bars represent the arithmetic mean, and the asterisks indicate the mean of the measurements that are significantly different according to a Student’s *t*-test with *p* < 0.05.

**Figure 6 plants-11-01526-f006:**
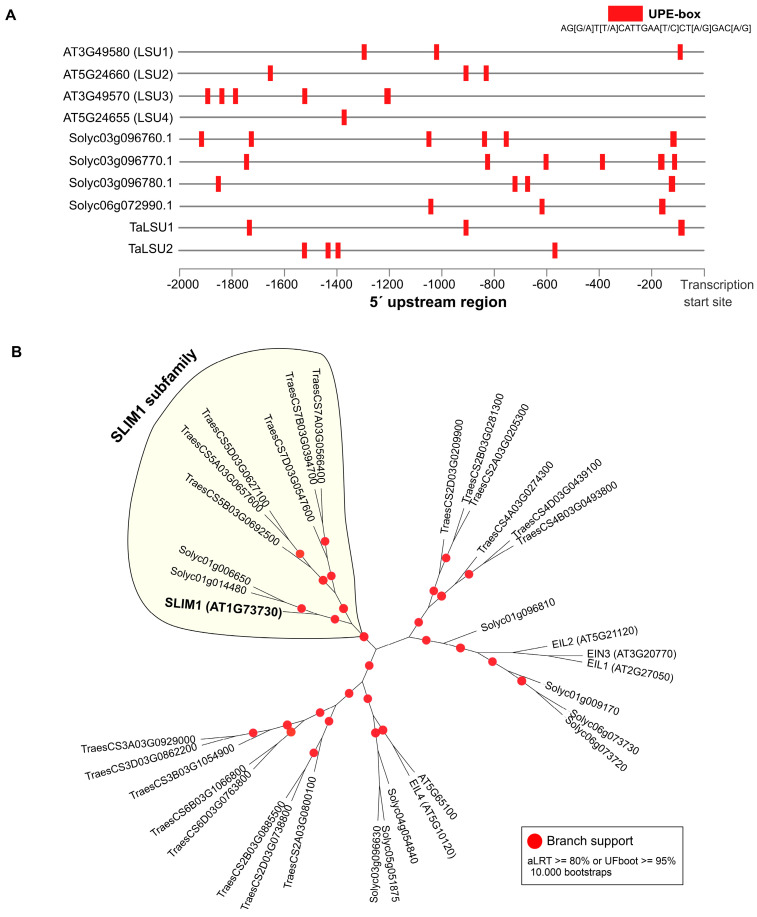
Distribution across promoters of the Arabidopsis, tomato and wheat *LSU* genes of the UPE-box element involved in S response. The promoter region was defined as the 2-kb sequence upstream of the transcription start site (based on Ensembl Plants v52) of each *LSU* gene (**A**). Phylogenetic analysis shows that ortholog genes of the Arabidopsis *SLIM1* are present in tomato and wheat genomes (**B**). SLIM1 is a well-known transcription factor that binds to the UPE-box elements [19,44].

## Data Availability

Not applicable.

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
