# Peer review of "Evolutionary and Gene Expression Analyses Reveal New Insights into the Role of LSU Gene-Family in Plant Responses to Sulfate-Deficiency"

_plants, 2022, doi:10.3390/plants11121526_

Round 1
Reviewer 1 Report
The resubmitted manuscript by Uribe et al. after removing the part of lsu2 mutant analyses
is definitely more coherent now. The authors have added additional data concerning the evolutional analyses of all members of the EIL family, and we identified SLIM1 orthologs in wheat and tomato. They also show the positions of UPE-box cis-acting element in the promoters of LSUs in Arabidopsis, potato, and wheat. However, I noticed some discrepancies between the original results presented by Wawrzyńska et all 2010 (doi:10.1093/jxb/erp356) and presented here for UPE-boxes position in Arabidopsis LSUs promoters. The authors did not show all positions for UPE-box for LSU2 and LSU3 (-160 bp and -163 bp, respectively). Please correct!
Author Response
Thank you very much for the positive and constructive comments.
POINT-BY-POINT RESPONSES TO REVIEWER 1:
- “However, I noticed some discrepancies between the original results presented by WawrzyÅ„ska et all 2010 (doi:10.1093/jxb/erp356) and presented here for UPE-boxes position in Arabidopsis LSUs promoters. The authors did not show all positions for UPE-box for LSU2 and LSU3 (-160 bp and -163 bp, respectively)”
RESPONSE: Thank you for noticing these potential discrepancies. In the paper of WawrzyÅ„ska et al 2010, the positions of UPE-boxes are indicated with respect to ATG (not the transcription start site) and the first UPE-boxes (-160 bp and -163 bp) are within the 5`UTR region according to the updated version of LSU gene models (Ensembl plants version 52). As indicated in Figure 6A and the main text, we focused our analysis on the promoter region 2 Kb upstream of the transcription start site. Therefore, we have not included the suggested UPE-boxes for LSU2 and LSU3 genes because these boxes belong to the 5’ UTR region (not the promoter region).
Reviewer 2 Report
Dear Authors,
I had an opportunity to review manuscript entilted „Evolutionary and gene expression analyses reveal new insights into the role of LSU gene-family in plant responses to sulfate-deficiency” resubmitted to the Plants MDPI Journal.
Authors concentrated on LSU gene-family in plant response to sulfate-deficiency.
However LSUs have arisen as „center”” in protein-protein interaction networks, in which they play importnat roles in the response to abiotic and biotic stress, most of knowledge on LSU genomic organization, expression and function comes from studies in Arabidopsis and also tobacco, while little is known about the LSU genes and evolution of this family in general in land plants.
Moreover, Authors analyzed the expression of LSU genes in one representative species of each phylogenetic group (wheat, tomato, and Arabidopsis) and found a conserved response to S-deficiency, suggesting that these genes might play a key role in S-stress responses. Furthermore, their results indicate that LSU genes belong to Spermatophyta-specific gene family and their response to S-deficiency is conserved in angiosperms.
Introduction provided sufficient background for obtained analyses, espacially interesting seems to be plant-microbe interaction aspect, despite of it I ask what was Authors hypothesis ?
Results are descibed in detail, but unfortunately only to the line 403 (??), after line 403 problems are starinting – First of all, the reader find that Figure 3 C was cut short in current form – Please, correct it;
Moreover, Figure 4 – in figure captions and in text citation we can find figure 4A and B – in the manuscript only part A was added -Please, correct it;
Furthermore, in Figure 5- the reader are able to see only part A (also in „cut short” version) , part B was not added;
Figure 7 (I guess) is completely cut short – the figure caption is lost, this figure is not cited in the manuscript;
Author Response
POINT-BY-POINT RESPONSES TO REVIEWER 2:
- “I ask what was Authors hypothesis?”
RESPONSE: The main goal of this work is to understand the evolutionary history of LSU gene family in plants and to get new insights into their possible role in S-deficiency response. The evolutionary conserved response to this nutritional stress would suggest an important functional role of this unknown protein family for S-responses. Therefore, our main hypothesis is that LSU genes are evolutionary conserved in angiosperm and sharing common regulatory elements of the S-deficiency response. The rationale of this work is included in the last paragraph of the introduction.
- “Results are descibed in detail, but unfortunately only to the line 403 (??), after line 403 problems are starinting – First of all, the reader find that Figure 3 C was cut short in current form – Please, correct it;
Moreover, Figure 4 – in figure captions and in text citation we can find figure 4A and B – in the manuscript only part A was added -Please, correct it;
Furthermore, in Figure 5- the reader are able to see only part A (also in „cut short” version) , part B was not added;
Figure 7 (I guess) is completely cut short – the figure caption is lost, this figure is not cited in the manuscript;
RESPONSE: We apologize for the confusion generated. The pdf file contained the highlighted changes with respect to the first submission of the manuscript. We have removed the highlighted changes in the pdf file to avoid this confusion, and all the mentioned errors have been corrected.
Round 2
Reviewer 2 Report
Authors significantly improved manuscript, especially the quality of results presentation- figures as well as figure captions were corrected;
Moreover, the research hypothesis was celarly explained;
Therefore, in my opinion the manuscript can be publicated in current form,
Sincerely
Author Response
Thank you very much for reviewing our work and for the positive comments.
This manuscript is a resubmission of an earlier submission. The following is a list of the peer review reports and author responses from that submission.
Round 1
Reviewer 1 Report
The manuscript by Uribe et al. reports on the origin and evolution of LSU-gene family in plants. The authors identified 270 LSU family members among 134 plant species. This part of the paper is impressive. We can find out that the evolutional appearance of LSUs is relatively recent compared to sulfur assimilation genes present already in Viridiplantae. From these analyses also a bigger functional divergence of LSUs within monocotyledon species is suggested. The article is based on recent references, most of them published after 2010 and many during last 5 years.
Though at first glance the paper looks encouraging I don’t think it is suitable for Plants in its current form. I would strongly advise dividing this paper into two papers. The first part should contain only evolutionary studies of LSU genes while the second paper should contain the analyses of lsu2 mutant, because they poorly connect with evolutionary studies. However, I think that this second part is based on preliminary results and the studies should be more detailed.
Just a small remark concerning the evolutionary part of this study. Because the authors found that in wheat, tomato, and Arabidopsis LSU genes are strongly up-regulated by sulfur-deficiency, I would suggest searching for UPE-box cis-acting element in their promoters. Especially, that these genes are so conserved in sulfur responses as presented in Fig 5. This would greatly improve the manuscript as it was already proven for Arabidopsis and tobacco that UPE-box is the element responsible for sulfur-deficiency-dependent upregulation of genes. It is interesting whether such an element is also conserved among plant species. Additionally, the authors might consider searching for SLIM1 homologs, as it was proven that SLIM1 is a transcription factor binding to UPE-box and responsible for LSUs increased expression. This might shed the light if the mechanism of up-regulation in sulfur deficiency is also conserved throughout the plant species.
As for the part concerning analyses of lsu2 mutant:
- I would rather suggest measuring the FW of plants than the shoot area. Is this phenotype always visible, ie. also at later stages of plant development?
- What about the flowering time?
- I am also surprised by the fact that Arabidopsis plants grown in sulfur deficiency (shown in Fig 6A) do not accumulate anthocyanins. It is a common protective mechanism in plants grown in stress conditions.
- How do the authors explain the changes in other gene expressions, especially in the ‘OAS cluster’ genes? How do LSUs affect transcription, do they interact with some transcription factors?
- And how can LSUs possibly take part in the regulation of internal sulfate levels? What’s the mechanism behind it?
- Lastly, I have an objection if lsu2 is a real knock-out. The LSU2 primers used in qRT-PCR frame the T-DNA insertion site so it is obvious they do not produce any product. However, the T-DNA insertion is in the middle of the coding region. We do not know if partial LSU2 protein is not produced. Such truncated LSU2 protein would still contain ‘motif 4’ and ‘motif 2’, which might still exert some functions. The simplest way would be to clone such partial LSU2 protein and check its interaction abilities (with other LSUs or other partners) by using yeast two-hybrid or pull-down assays. It is quite possible that in lsu2 mutant instead loss-of-function we might observe gain-of function. I’ am very surprised that the authors even saw the phenotypic, transcriptomic, and metabolomics changes in this single mutant. They even claim themselves that they noticed compensatory effect with other members of the LSU family. Therefore the most elegant way to study the function of these proteins would be to study the knock-out of all four LSU genes to avoid compensation and false observations.
Finally, this paper needs English revision as sometimes it does not read smoothly. There are several apparent mistakes, eg . line 602 – ‘phytohormes’, line 633 - duplication of ‘the interaction’.
Therefore, in its current form, I do not recommend the publication of this article in Plants. I strongly suggest restricting this article to only evolutionary analyses of LSUs.
Reviewer 2 Report
The manuscript “Evolutionary and gene expression analyses reveal new insights into the role of LSU gene-family in plant responses to sulfate
deficiency” can be ideally divided into two distinct parts: i) the first reports an evolutionary analysis of the LSU gene family in plants; ii) the second reports an analysis of the expression of LSU genes in 3 representative plant species (Arabidopsis; tomato; wheat).
I really failed in understanding the rationale of this work. It seems the results of a merge between two independent analyses.
I would like to suggest to the Authors that the best way to valorize their data is to use the first part of the paper to write an opinion on the LSU genes in plants. The second part of the paper should be integrated with other results in order to better discuss the role of LSU2 also in relation to the finding previously reported in other papers (please, see: Luhua et al.(2013) Physiol Plant. 148, 322–333; Mukhtar et al. (2011). Science 333, 596–601).